# 4D trajectory prediction and conflict detection in terminal areas based on an improved convolutional network

Xin Ma [1]*, Linxin Zheng [2], Xikang Lu [2]

**1** Nanjing University of Aeronautics and Astronautics, Nanjing, China, **2** Civil Aviation Flight University of China, Sichuan, Guanghan, China

* maxin@cafuc.edu.cn

**Data Availability Statement:** All relevant data are within the paper and its Supporting Information files.

**Funding:** The author(s) received no specific funding for this work.

## Abstract

At present, the passenger traffic volume of civil aviation is gradually increasing, and the scale of the airline network is gradually expanding. In order to optimize the air traffic service mode more safely and scientifically, the International Civil Aviation Organization (ICAO) has proposed a new concept of trajectory based operation (TBO). According to the basic principle and core structure of TBO, aiming at improving the accuracy of Four Dimensional (4D) Trajectory prediction and the reliability of multi-trajectory flight conflict detection, the trajectory prediction model is established by using convolutional neural networks-bidirectional gated recurrent unit (CNN-BiGRU), and the conflict evaluation of prediction results is realized by using trajectory distance detection function. The simulation experiment shows that the simulation experiment is carried out by introducing the real automatic dependent surveillance-broadcast (ADS-B) historical track data in the terminal area of the busy airport. The experimental results are compared with the experimental results of the single long short-term memory (LSTM) model and the gated recurrent unit (GRU) model in the same data set. The results show that the CNN-BiGRU trajectory prediction model is superior to the comparison model in many evaluation indexes, and the conflict detection results can be evaluated for the future 800 seconds interval of the two trajectories.

## 1. Introduction

The terminal area is the airspace between the airport and the route. It generally refers to the distance from the airport base point within 50 km, the height below 6600 m (not included), and above the lowest altitude level [1]. Due to the heavy traffic and limited space in the terminal area, the risk of flight conflicts is high. Therefore, accurate prediction and conflict detection of flights in the terminal area is extremely important, and it will contribute to ensure aviation safety, improve transportation efficiency and reduce operating costs [2]. The 4D (Four Dimensional) trajectory refers to the orderly collection of position and time experienced by the aircraft in the three-dimensional space during the entire process from takeoff to landing [3], and time is the fourth dimension. Accurate prediction of 4D trajectory is a key step in flight control and decision-making in the terminal area, which can improve the operation efficiency of the

**Competing interests:** The authors have declared that no competing interests exist.

terminal area. Conflict detection technology refers to monitoring whether there is a conflict with other aircraft or ground obstacles in the flight state of the aircraft, so as to take corresponding measures to ensure aviation safety. Effective flight conflict detection is an essential key step in air traffic management [4].

With the continuous updating of communication, navigation, surveillance and airborne equipment, as well as the development of computer technology and data mining methods, more and more data-driven methods have been proposed. Deep learning is an algorithm that automatically learns and predicts from a large amount of data [5]. Among them, long short-term memory (LSTM) and gated recurrent unit (GRU) are commonly used algorithms to capture the long-term dependence of time series data. Through the learning of historical flight data, the future trajectory can be predicted through the aircraft trajectory operation mode. Based on deep learning, Ma [6] et al. proposed a new 4D trajectory prediction composite structure. Shafienya [7] et al. used Monte Carlo dropout (MC-Dropout) to enhance the predictability of the model, and reduced the 4D trajectory prediction error by an average of 21%. Zhao Ziyu [8] extracted the flight characteristics and typical route change points of the global segment through data mining, and proposed an aircraft intention inference model based on Bayesian theory, which has better prediction accuracy and robustness. Zhang [9] et al. combined the LSTM model of deep learning with the extreme Gradient Boosting (XGBoost) model to enhance the generalization ability and practicability of the model. Based on accurate trajectory prediction technology, the conflict detection methods used by scholars at home and abroad are usually divided into geometric determination and probability analysis. Ding Songbin [10] and other trajectory prediction models based on machine learning, combined with the conflict warning function, evaluate the safety of a given trajectory. Wang Zekun [11] et al. optimized the traditional speed obstacle method model, and independently selected the release strategy for conflict detection in flight. Hao Siqi [12] proposed conflict detection, conflict probability quantification and risk assessment methods for aircraft in potential space. Madar [13] et al. combined trajectory clustering, classification-based supervised learning and probability modeling to calculate the probability of conflict, and promoted the detection and resolution of potential threats in the terminal airspace.

At present, there are still two main challenges in 4D trajectory prediction in terminal area: data quality and model robustness. Missing data and noise are one of the problems, while the other is the adaptability of the model in different situations. In addition, effective terminal area flight conflict detection needs to be analyzed in combination with aircraft operation. This paper proposes a prediction model (hereinafter referred to as prediction model) and geometric conflict detection method based on convolutional neural networks-bidirectional gated recurrent unit (CNN-BiGRU). The aim is to improve the safety and operational efficiency of the aircraft in the terminal area by comprehensively considering various factors such as data quality, model robustness and conflict detection.

## 2. Model establishment

### 2.1 Data preprocessing

Automatic dependent surveillance-broadcast (ADS-B) is a technology that connects aircrafts and ground stations. It connects satellites, aircrafts and ground stations to form an integrated system involving three levels: space, air and ground. It is widely used in civil aviation and public safety and other fields by sending ADS-B information to the outside world to report the current flight parameters of the aircraft and the specific location information of the aircraft. The ADS-B training data set is adopted in this paper, but because of the noise value and missing value of the data, data analysis and preprocessing are needed to improve the mathematical expression.

Assuming that T is a set of historical trajectories containing n historical trajectories, denoted by:

$$T = \{T_1, T_2, \ldots, T_n\} \tag{1}$$

Tk is the k th trajectory in the track set. Assuming that each trajectory contains n track points, there is:

$$T_k = \{m_{k1}, m_{k2}, \ldots, m_{kn}\} \tag{2}$$

$m_{ki}$ is the i th track point in Tk. If each track point contains n features, there is:

$$m_{ki} = \{r_{ki1}, r_{ki2}, \ldots, r_{kin}\} \tag{3}$$

$r_{kij}$ is the j th feature of track point $m_{ki}$.

The characteristics of each collected historical track are shown in Table 1.

In order to make accurate 4D trajectory prediction, the preprocessing of ADS-B data is extremely important. To solve this problem, the interpolation of ADS-B data using cubic splines can improve the accuracy of 4D trajectory prediction. Cubic spline interpolation is a commonly used data interpolation method, which can fit new data points by the curve between known data points. In ADS-B data cleaning and interpolation, not only noise points need to be removed, but also missing data points need to be completed. Firstly, it is determined whether the point is a noise point by detecting whether the distance of the data point on the curve is too large. If it is, it is removed to avoid its interference with the prediction results. For missing data points, a more complete data set can be obtained by interpolation.

Assuming $f(x)$ is a quadratic continuous differentiable function on the interval [a, b]. The interval [a, b] is divided into n intervals:

$$[(x_0, x_1), (x_1, x_2), \ldots, (x_{n-1}, x_n)] \tag{4}$$

There are n+1 points, and $x_0 = a$, $x_n = b$, then:

$$T(x) = \begin{cases} T_1(x), x \in [x_1, x_2] \\ \ldots \\ T_i(x), x \in [x_i, x_{i+1}] \\ \ldots \\ T_n(x), x \in [x_n, x_{n+1}] \end{cases} \tag{5}$$

If $T(x_i)$ satisfies the following four conditions:

**Table 1. Illustration of historical trajectory features.**

|  | trait name | track point |
|---|---|---|
| $r_{ki1}$ | time | 2022-12-07 05:38:10 |
| $r_{ki2}$ | longitude (˚) | 140.3917 |
| $r_{ki3}$ | latitude (˚) | 35.7421 |
| $r_{ki4}$ | altitude (ft) | 12528 |
| $r_{ki5}$ | velocity (kt) | 321 |
| $r_{ki6}$ | heading angle (˚) | 65 |

Note: $r_{kij}$ is the j th feature of k track at time i in ADS-B data; 1 ft = 0.304 8 m; 1 kt = 1.852km / h.

**Table 2. Pre-processing track data.**

| time | latitude (°) | longitude (°) | altitude (ft) | velocity (kt) | heading angle (°) |
|---|---|---|---|---|---|
| 08:30:04 | 21.9323 | 113.2976 | 1625 | 134 | 096 |
| 08:30:10 | 21.9351 | 113.3006 | 1550 | 134 | 097 |
| 08:30:15 | 21.9368 | 113.3025 | 1500 | 134 | 097 |
| 08:30:41 | 21.9472 | 113.3134 | 1200 | 135 | 097 |
| 08:34:08 | 22.0089 | 113.3765 | 0 | 6 | 016 |

1. When calculating the existing trajectory position point data, the interpolation results should be equal to the original data, then there is:

$$T(x_i) = f(x_i), i = 1, 2, \ldots, n+1 \tag{6}$$

2. When $T(x)$ is computed in $[x_i, x_{i+1}](i = 1,2,\ldots,n+1)$, the polynomial or zero polynomial can be constrained to be no higher than three times:

$$T(x) = a_i + b_i x + c_i x^2 + d_i x^3 \tag{7}$$

3. $T(x)$ is twice continuously differentiable, i.e.

$$\lim_{x \to x_i} T'(x) = T'(x_i) = m_i, i = 1, 2, \ldots n-1 \tag{8}$$

$$\lim_{x \to x_i} T''(x) = T''(x_i) = m_i, i = 1, 2, \ldots n-1 \tag{9}$$

4. Because the track data is not negative, if the calculation result is negative, it is equal to the minimum value in the interval, there is:

$$T(x) = \min[x_i, x_{i+1}], if \ T(x) < 0 \tag{10}$$

The data sample before pre-processing is shown in Table 2.

A sample of the pre-processed data is shown in Table 3.

In summary, $T(x)$ is the cubic spline interpolation model of the corresponding track data with $f(x)$. When performing 4D trajectory prediction, the processed ADS-B data can more accurately reflect the real state of the flight and provide a more accurate data base, thereby improving the accuracy of the prediction results.

**Table 3. Post-processing track data.**

| time | latitude (°) | longitude (°) | altitude (ft) | velocity (kt) | heading angle (°) |
|---|---|---|---|---|---|
| 08:30:05 | 21.9323 | 113.2976 | 1625 | 134 | 096 |
| 08:30:10 | 21.9328 | 113.2981 | 1612 | 134 | 097 |
| 08:30:15 | 21.9333 | 113.2987 | 1599 | 134 | 097 |
| 08:30:20 | 21.9337 | 113.2992 | 1586 | 134 | 097 |
| 08:30:25 | 21.9342 | 113.2997 | 1573 | 134 | 096 |

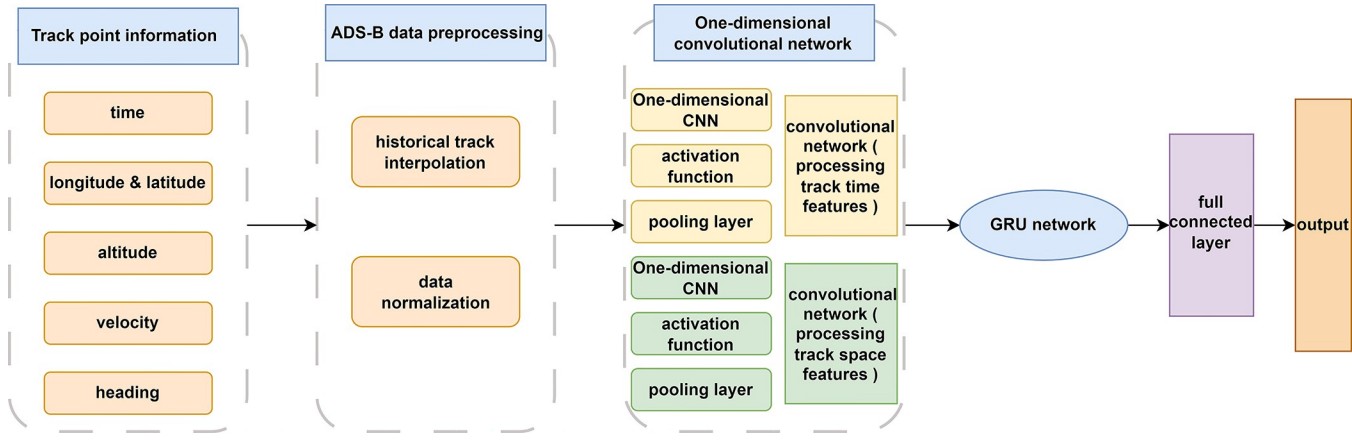

**Fig 1. CNN-BIGRU model structure.**

## 2.2 Model prediction

Accurate trajectory prediction is the basis of detecting flight conflicts. Accurate 4D trajectory prediction model can effectively improve the accuracy and reliability of prediction. Previous studies have shown that 4D trajectory prediction needs to be further optimized and condensed, especially in the aspects of historical track data volume, time series characteristics, calculation logic and prediction time. Therefore, a combined model is proposed to improve the comprehensiveness and accuracy of 4D trajectory prediction. The known N track points: $\{m_{ki(N-1)}, \cdots, m_{ki-1}, m_{k-1}\}$ are introduced, and the CNN-BiGRU model is used to predict the future track points N': $\{m_{ki+1}, \cdots m_{ki+2}, m_{ki+n}\}$. With the goal of accurate prediction, the prediction results of spatial-temporal characteristics and multi-source and multi-level characteristics are realized. At the same time, the reliability of the prediction model is analyzed and evaluated from the aspects of predictable duration and time series characteristic index.

The trajectory prediction process of the prediction model is divided into three steps: preprocessing the track position point information, identifying the input effective track data, and training and outputting the prediction results with a two-way gated network. The data preprocessing stage avoids the influence of ADS-B data error on the prediction accuracy. One-dimensional convolution extracts the spatial features of track position points and generates a 4D track data spatial feature sequence. The bidirectional gated loop part processes the complex nonlinear relationship in the data sequence, extracts the time dimension features of the track data, and achieves high-precision 4D trajectory prediction results. The structure of the CNN-BiGRU model is shown in Fig 1.

Through analysis and pre-processing, the characteristics of the track position point of the effective track data at time i are:

$$T_i = \{t, lon, lat, alt, vel, h\} \tag{11}$$

Among them, t, lon, lat, alt, vel and h represent the time, longitude, latitude, altitude, speed and heading of track T at time i, respectively. The input data format is time series tensor. In order to facilitate the convolution operation of the convolutional network and improve the prediction accuracy, the time step is set to 6, and the next track point data is predicted by using the first 6 continuous track point information.

In terms of activation function, the use of Relu can simplify the calculation and reduce the cost. The Relu activation function formula is:

$$f(x) = relu(x) = \begin{cases} x, x \geq 0 \\ 0, x < 0 \end{cases} \tag{12}$$

Through the Relu function, if the input x is greater than or equal to 0, the output is x; if it is less than 0, the output is 0.

After the processing of the pooling layer, the number of data parameters is reduced, the error of the convolution layer is corrected, and the stability of the computational efficiency and spatial feature extraction ability is improved. At the same time, in order to prevent over-fitting, each bidirectional gated recurrent network adds a dropout layer. The dropout layer randomly resets part of the weight or output of the hidden layer to zero to reduce the interdependence between neural network nodes, thereby avoiding over-fitting. Finally, the output of the second bidirectional gated recurrent network will be passed to the fully connected layer to integrate all local features and output the calculation results of the time, longitude, latitude and altitude of the aircraft at a future time.

## 2.3 Conflict detection function

High-precision trajectory prediction technology is the basis of flight conflict detection. Accurate conflict detection is based on accurate trajectory prediction technology [14]. It effectively reduces the workload of controllers and improves the intelligence of air traffic. Especially in the terminal area, adjusting the horizontal and vertical intervals of the aircraft is a common means of control deployment. Through high-precision trajectory prediction technology, the trajectory of the aircraft can be predicted more accurately, and the occurrence of conflicts can be avoided, thus effectively ensuring the safe operation of the aircraft. In the flight phase, the flight interval is the minimum safe distance that should be maintained between aircrafts to prevent flight conflicts and ensure flight safety, as well as to improve the utilization of flight space and time. The flight interval includes vertical interval and horizontal interval [4]. In the flight process, in order to further ensure flight safety and reduce the risk of conflict, a safe area is constructed for the flight mission implementation phase: the horizontal safety interval $H_c$ of the aircraft and the vertical safety interval $V_c$ together form the flight protection area of the aircraft. There are many kinds of aircraft protection area models, including cylindrical and rectangular shapes. Because the cylindrical space occupancy rate is relatively low, the proposed cylindrical flight protection zone model has a radius of $H_c$ and a height of $2V_c$, as shown in Fig 2.

The aircraft conflict detection function can be expressed as follows:

$$I = \sum_{i=1}^{n} [d_i^h(p_i(A), P_i(B)) < \eta\delta^h \ and \ d_i^v(p_i(A), P_i(B)) < \eta\delta^v] \tag{13}$$

$d_i^h$ and $d_i^v$ are horizontal and vertical distance functions, respectively. $p_i(A)$ and $p_i(B)$ are the latitude and longitude positions of aircraft A and B at time i; $\delta^h$ and $\delta^v$ are the horizontal and vertical interval standards set by the terminal area control center, and $\eta$ is the coefficient set to meet different early warning needs. In addition, the detection function I is defined as: if the horizontal distance and vertical distance of the two aircrafts do not meet the set control interval, the function value is 1 and the conflict warning is performed; otherwise, the value is 0, and the two aircrafts can maintain a safe interval.

By fusing the trajectory distance detection function, the 4D flight trajectory prediction is combined with the safety assessment between two given trajectories. If the short-term

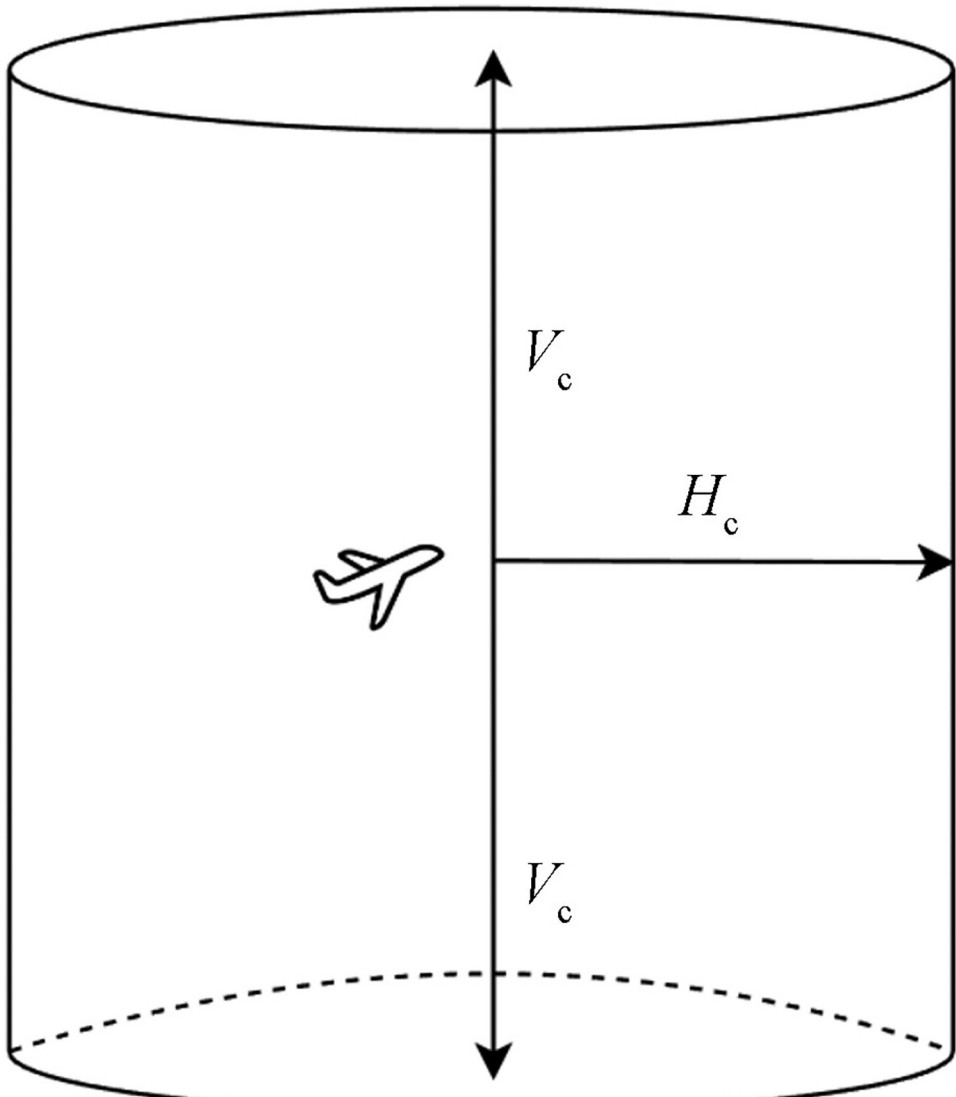

**Fig 2. Flight interval protection zone.**

trajectory distance calculation result is less than the safety interval threshold, an alarm prompts appropriate measures to improve safety.

## 3. Simulation experiment

In order to better verify the trajectory prediction and conflict warning function proposed in this paper, real historical trajectories in a busy terminal area are used. According to the airspace altitude area division (range: 900m-4,500m) specified in the terminal area, some approach and approach trajectories of an aircraft are selected for 4D trajectory prediction experiment analysis. The whole process of the simulation experiment is shown in Fig 3.

In order to enhance the prediction capability and improve the computational efficiency while reducing the prediction error and noise, the track data should be normalized before

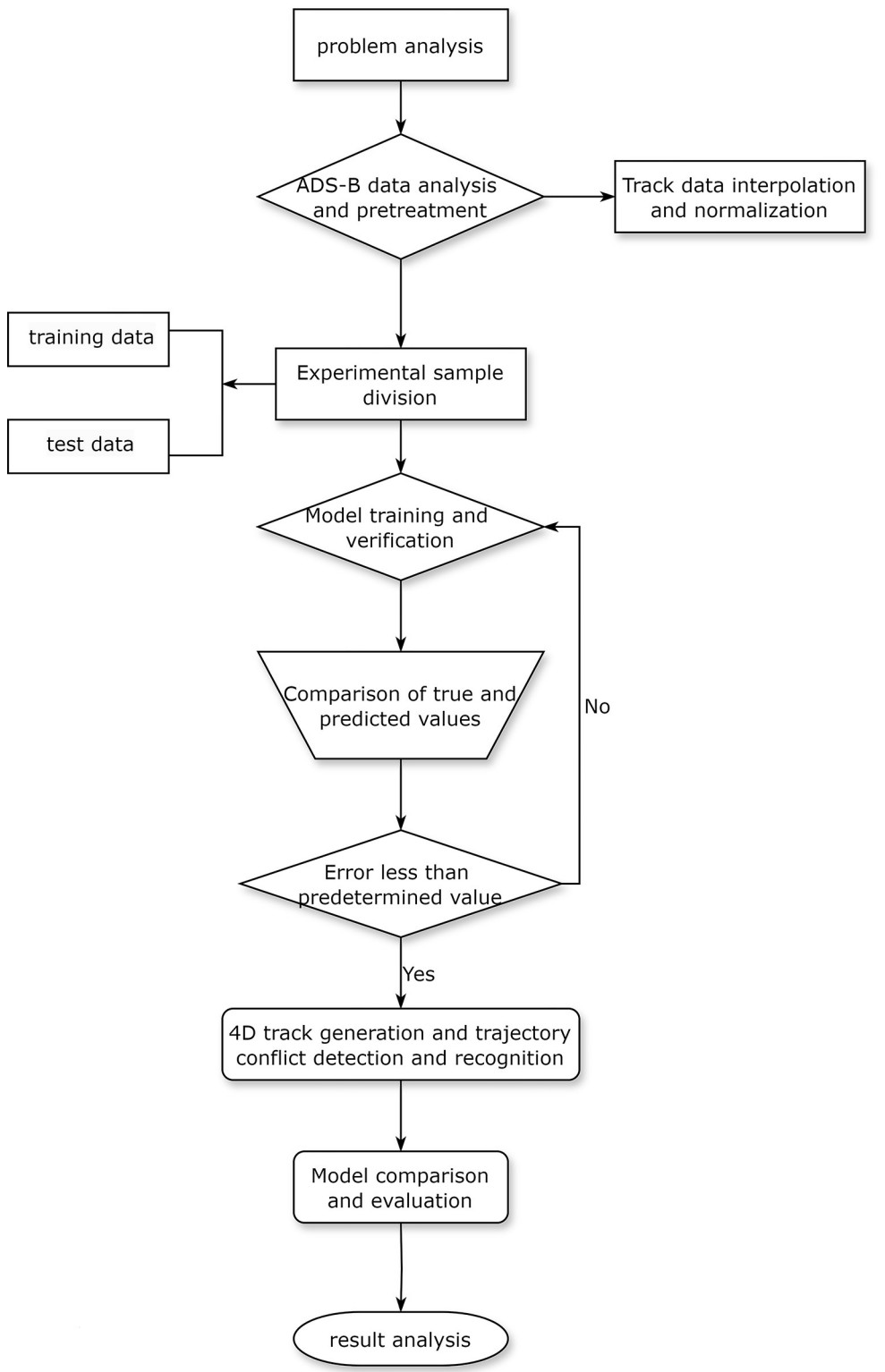

**Fig 3. Simulation experiment process.**

being imported into the training model:

$$N = \frac{X - Min}{Max - Min} \qquad (14)$$

X is the original trajectory sample data, Max and Min denote the maximum and minimum values of the sample, respectively, and N is the normalized sample.

Root Mean Squared Error (RMSE) and Mean Absolute Percentage Error (MAPE) are the most commonly used evaluation indicators in regression problems [15]. RMSE is the expected value of the square of the difference between the predicted result and the actual target, and then the square root operation is taken; MAPE is a process of comparing with the original data, considering the ratio of error to actual value. By using the above indicators to measure the error, the effectiveness of the CNN-BiGRU model proposed in this paper is evaluated. The calculation formula of the above indicators is as follows:

$$RMSE = \sqrt{\frac{1}{m}\sum_{i=1}^{m}(p_i - a_i)^2} \qquad (15)$$

$$MAPE = \frac{1}{m}\sum_{i=1}^{m}\left|\frac{p_i - a_i}{p_i}\right| \times 100\% \qquad (16)$$

$p_i$ is the predicted value of the i th track point, and $a_i$ is the real value of the i th track point. When the calculated value of the index is smaller, it is proved that the smaller the error between the predicted value and the real value, the better the model prediction result. In order to improve the ability of training and prediction, the pre-processed track information in the simulation experiment is used as feature data and label data. Then the feature data and label data are divided into training set and test set, accounting for 70% and 30% of the total data set, respectively. In addition, 10% of the training set of the data set was selected as the validation set to verify the model. In order to reduce the prediction error, a single-step prediction input data method is proposed to construct the trajectory sample, as shown in Fig 4.

Fig 4 shows the trajectory samples divided in the simulation experiment. Each row represents a time step, and each column represents the characteristics of the data. The experiment starts from the first trajectory position point, and selects the time, longitude, latitude, height, speed and heading angle of the six track position points one by one in time order. Based on this, the time, longitude, latitude and height of the next trajectory position point are predicted. It can be seen that the trajectory sample is divided into a 6 * 6 matrix.

Secondly, the actual track data and the calculation results of the prediction model are plotted to compare the accuracy of the prediction results. At this stage, the LSTM model and the GRU model are used as comparison models, and the same data set is used for experiments [6]. Among them, Fig 5 is a two-dimensional diagram of multi-model prediction results and actual latitude and longitude coordinate trajectories. The CNN-BiGRU model has obvious advantages over the other two methods in terms of track data error, prediction accuracy and curve fitting, and all kinds of deviation values are in a small state. Fig 6 shows the three-dimensional display of the multi-model predicted trajectory and the actual trajectory. After adding the elevation data, the elevation data prediction results of the CNN-BiGRU model are also more accurate. The reason for the above difference is that when the CNN-BiGRU model is used for trajectory prediction, the spatial and temporal characteristics of the track data can be processed in parallel, which has the advantages of higher accuracy and better fitting than other methods.

The prediction model proposed in this paper shows better performance in the longitude and latitude prediction of 4D trajectory. Compared with the other two comparison models,

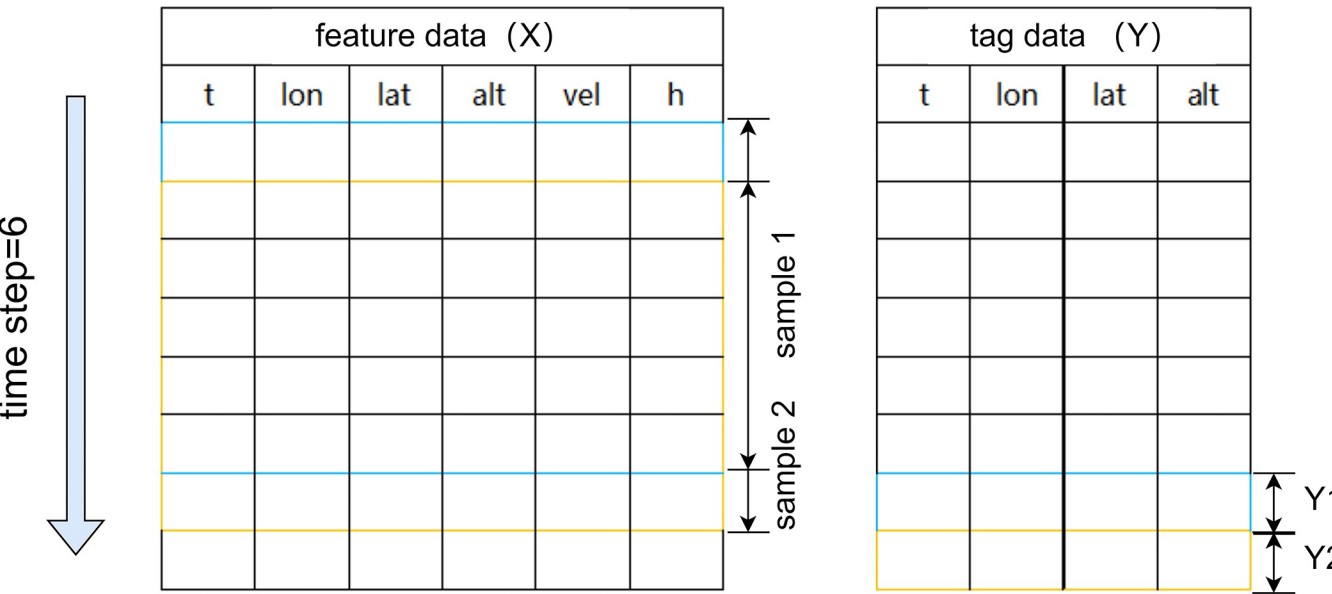

**Fig 4. Trajectory sample split chart.**

the CNN-BiGRU model has the smallest prediction error, followed by the GRU model, and finally the LSTM model. Because the convolution part is introduced into the proposed model, the spatial and temporal characteristics of the track data can be processed. Therefore, it has advantages in solving the problems of lack of trajectory prediction dimension and insufficient prediction accuracy. After optimization, the prediction model can effectively avoid over-fitting and further improve the accuracy of 4D trajectory prediction. In addition, the model has a two-way gated loop unit, which can process the track data in a two-way one-by-one time dimension. Therefore, the prediction results can cover the complete historical and future information, and improve the prediction accuracy and length.

According to the comparative analysis of the predicted trajectory and the actual trajectory, the errors of the four characteristics of time, longitude, latitude and height in the single-step prediction are statistically analyzed, and the results are shown in Table 4. It can be seen that compared with GRU and LSTM models, the error of the prediction model is small, and the evaluation indexes are better than the former two. At the same time, the evaluation index of GRU model for single feature is better than that of LSTM model. It can be seen that the prediction model shows higher accuracy in 4D trajectory prediction, and the deviation between the prediction results and the actual data is smaller. This proves that the prediction performance of the prediction model is more stable in dealing with time series data.

## 4. Example verification

In order to consider the conflict recognition performance of the model in the actual operating environment, the real data of another aircraft in the same terminal area is first introduced for track prediction. At the same time, a new trajectory is generated to realize the distance interval calculation and conflict detection between the two aircrafts. For simplification of calculation, it is assumed that the two aircrafts operate at the same time. Among the above prediction models, the CNN-BiGRU model has the highest prediction accuracy, and the generated prediction trajectory can be used for interval calculation and potential conflict detection.

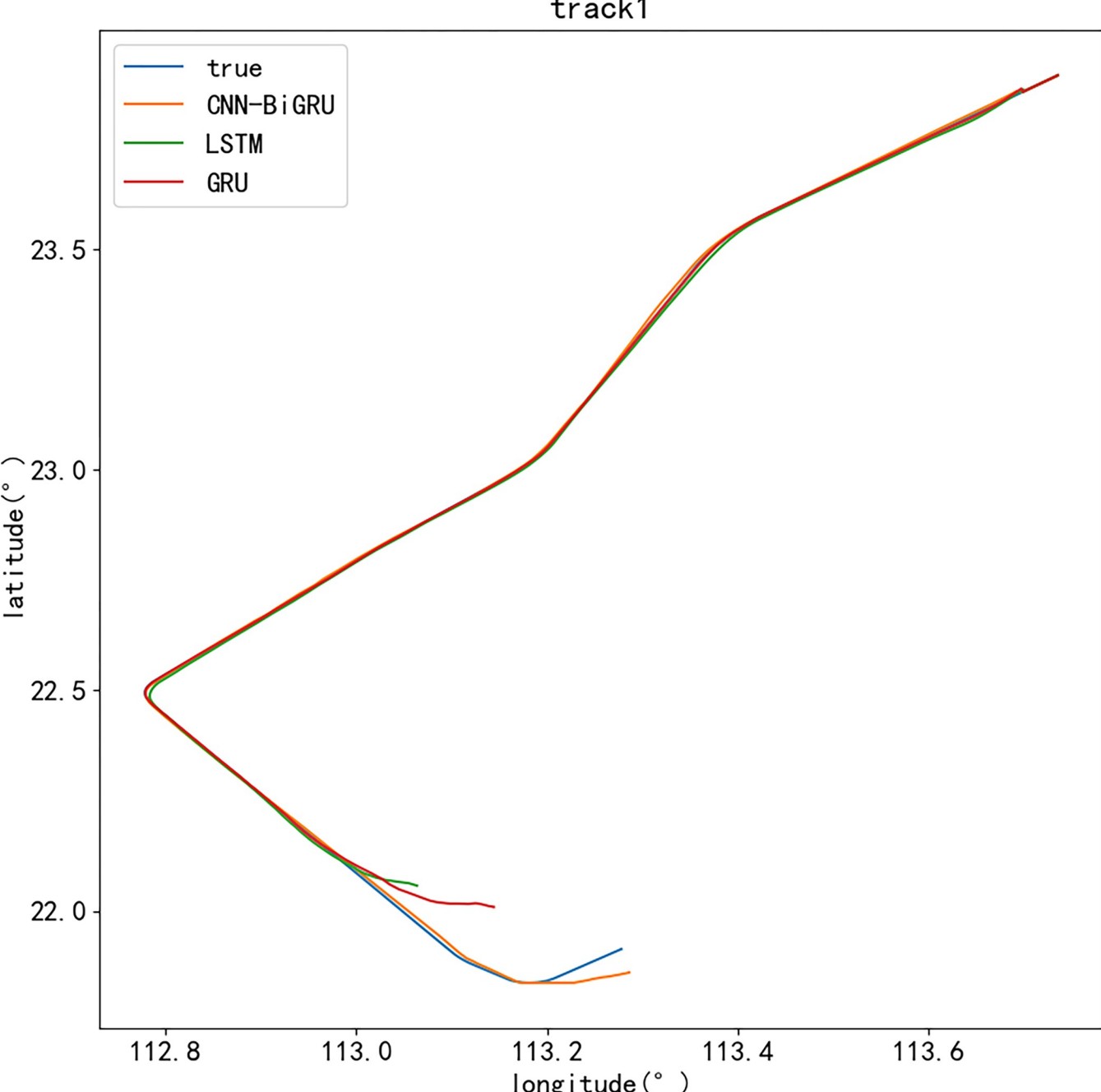

**Fig 5. Trajectories comparison of longitude and latitude.**

By checking the flight horizontal distance and vertical distance, the interval calculation is carried out to verify whether the conflict detection function can effectively identify the trajectory conflict. At the same time, the distance difference between the horizontal and vertical distance difference of the two aircraft trajectories generated by the prediction model and the real trajectory is compared to verify the reliability of the prediction model. According to the terminal area control operation center setting, the horizontal interval of the aircraft in the terminal area is 6 000 m, and the vertical interval is 300 m. The ratio coefficient is set to 1 to meet the

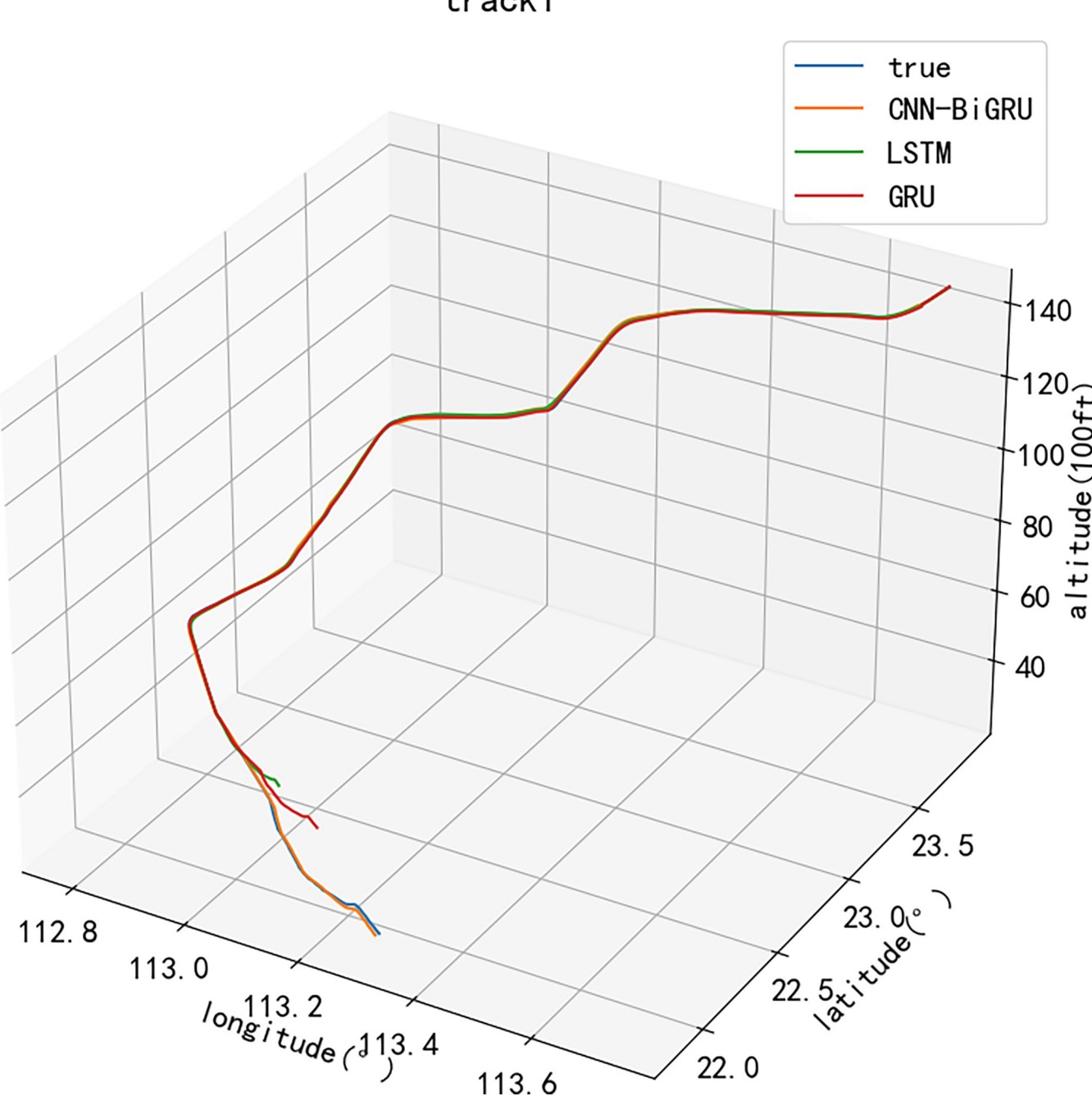

**Fig 6. Comparison of 3D trajectories.**

**Table 4. Comparison of total error of multi-prediction models.**

| evaluation index | CNN-BiGRU model | GRU model | LSTM model |
|---|---|---|---|
| RMSE | 1.004679 | 2.889301 | 4.165763 |
| MAPE | 0.028814 | 0.231659 | 0.332738 |

minimum early warning demand. Since the aircraft position data collected by ADS-B only contains the latitude and longitude information of each track point, it cannot be directly used for distance calculation. In order to meet the needs of conflict detection, the spherical cosine theorem is used to calculate the actual horizontal distance from the latitude and longitude data of any two track points. The difference of vertical distance is the flight height difference $|r_{1i4}-r_{2i4}|$ of the two aircrafts at the same time. The specific steps are as follows:

1. First, the latitude and longitude are converted into radians, the longitude is multiplied by $\frac{\pi}{180}$ and the latitude is also multiplied by $\frac{\pi}{180}$;

2. To calculate the Earth 's radius R, usually take 6,371.01 km;

3. The spherical angular distance is calculated according to the spherical cosine theorem:

$$\cos\Delta\sigma = \sin\varphi_1 \times \sin\varphi_2 \times \cos\varphi_1 \times \cos\varphi_2 \times \cos(\varphi_2 - \varphi_1) \tag{17}$$

$\Delta\sigma$ denotes the spherical angular distance between two points. $(\varphi_1\lambda_1)$ and $(\varphi_2\lambda_2)$ are the latitude and longitude coordinates of the two points to be calculated respectively.

4. Finally, the spherical angular distance is converted to the actual distance according to the radius of the sphere:

$$d = R \times \Delta\sigma \tag{18}$$

Therefore, the processed position data of the two aircrafts are used for interval calculation, and the detection function constructed by Eq (13) is used to test whether there is a potential conflict. Due to the large amount of track data, the 800 s closest to the two aircrafts is selected as the sample to output the vertical and horizontal distance comparison diagram. The experimental results are as follows:

Figs 7 and 8 show the calculation results of the vertical and horizontal distance difference between the predicted trajectory and the actual trajectory of the two planned simultaneous operation trajectories. As shown in the above figure, the curve of the predicted trajectory distance difference and the actual trajectory distance difference is basically fitted within 800 s, and the distance between the track points is accurately identified, which proves that the model proposed in this paper has good reliability. At the same time, according to the vertical and horizontal safety intervals in the terminal area set above, the conflict detection function can effectively detect the conflict situation. The two aircraft tracks generated by the prediction model have no flight conflict in the next 800 s to meet the requirements of safe operation in the terminal area.

## 5. Conclusion

According to the analysis of the calculation results of the simulation experiment and the comparative experiment, the following conclusions are drawn:

1. Through the interpolation preprocessing of ADS-B data, the spatial dimension and time dimension characteristics of 4D track data are integrated, and the CNN-BiGRU model is established to complete the high-precision prediction of 4D trajectory. By combining trajectory prediction with conflict detection, the conflict detection function is established, and the effective detection of early flight conflict is realized.

2. The aircraft position and track data in the airport terminal area are selected for experiments. The experiment proves that the CNN-BiGRU model can accurately predict the 4D

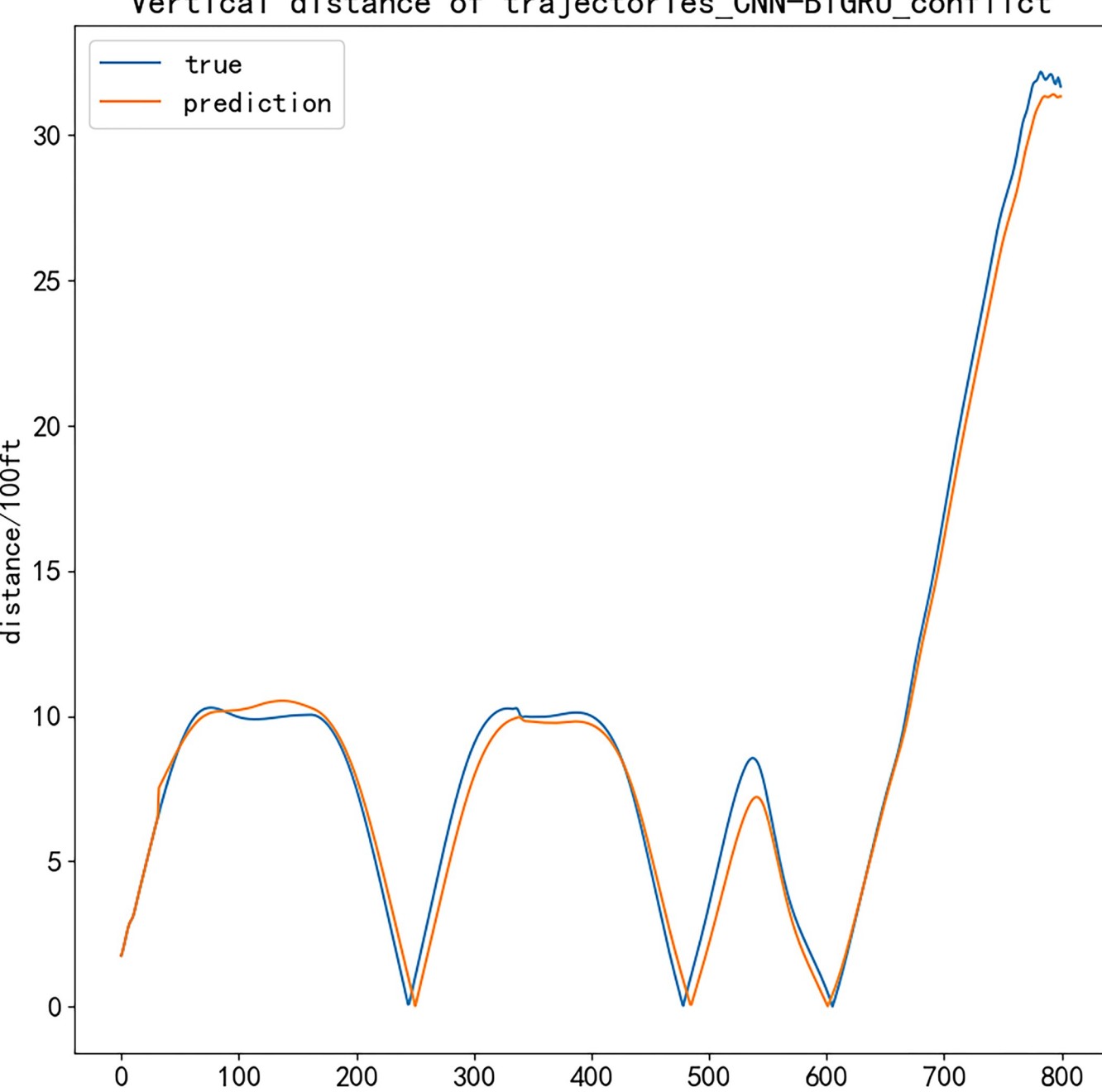

**Fig 7. Vertical distance of trajectories.**

trajectory, and can predict the conflict relationship and influence degree of the two trajectories. The prediction interval is increased to the next 800 seconds.

3. When the CNN-BiGRU model proposed in this paper is used to predict the 4D trajectory, the prediction error is greatly reduced compared with the GRU model and the LSTM model, which improves the accuracy of the 4D trajectory prediction and further meets the needs of actual operation.

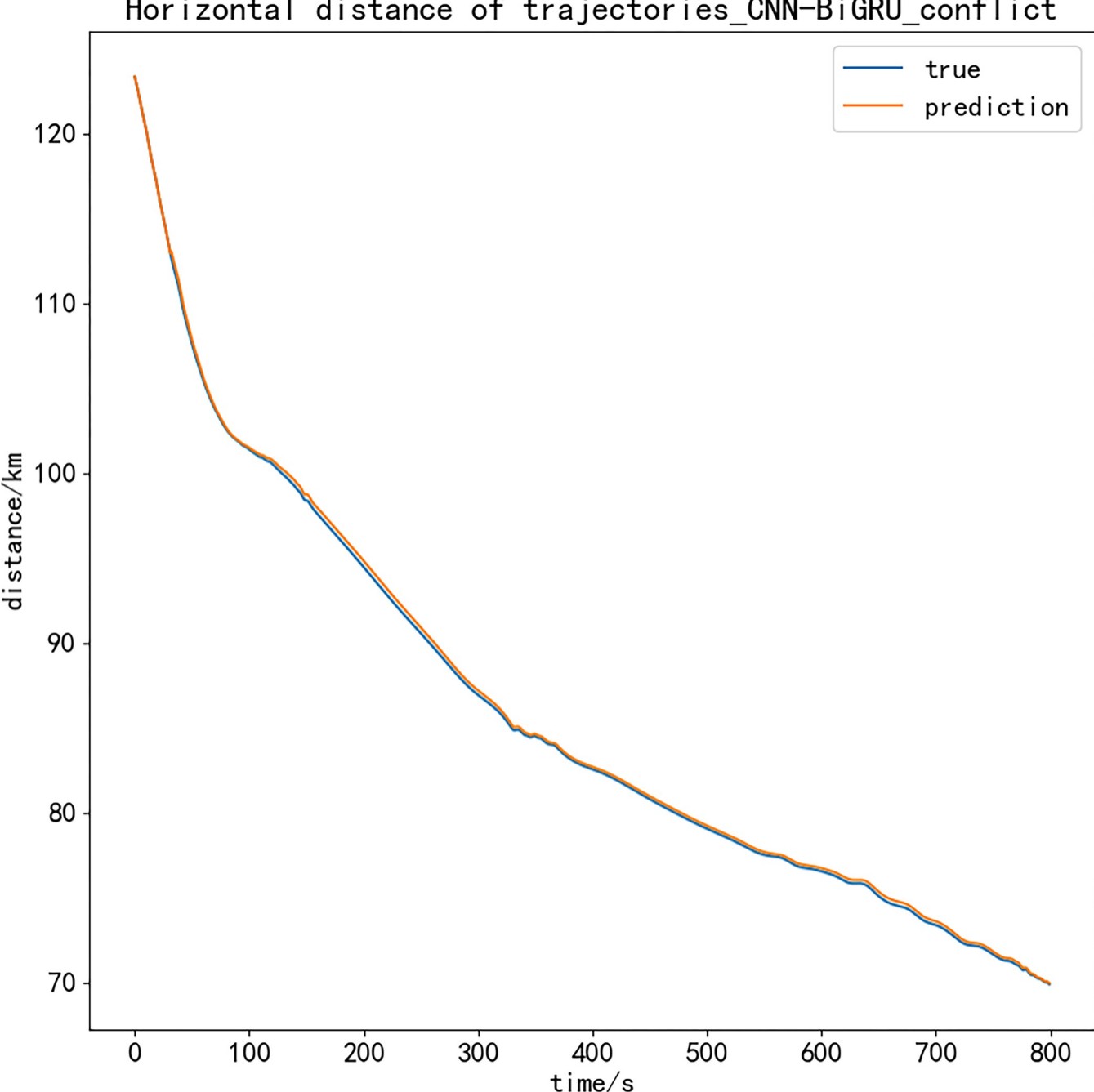

**Fig 8. Horizontal distance of trajectories.**

## Supporting information

**S1 File.**
(XLS)

**S2 File.**
(XLS)

## Author Contributions

**Resources:** Xikang Lu.

**Writing – original draft:** Xin Ma, Linxin Zheng, Xikang Lu.

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
