## [Decision Letter · Decision Letter 0]

9 Sep 2024

PONE-D-24-288194D trajectory prediction and conflict detection in terminal areas based on an improved convolutional networkPLOS ONE

Dear Dr. Zheng,

Thank you for submitting your manuscript to PLOS ONE. After careful consideration, we feel that it has merit but does not fully meet PLOS ONE’s publication criteria as it currently stands. Therefore, we invite you to submit a revised version of the manuscript that addresses the points raised during the review process.

We look forward to receiving your revised manuscript.

Kind regards,

Xiangjie Kong

Academic Editor

PLOS ONE

Journal Requirements:

Reviewers' comments:

Reviewer's Responses to Questions

**Comments to the Author**

1. Is the manuscript technically sound, and do the data support the conclusions?

Reviewer #1: Yes

Reviewer #2: Partly

2. Has the statistical analysis been performed appropriately and rigorously? 

Reviewer #1: Yes

Reviewer #2: N/A

3. Have the authors made all data underlying the findings in their manuscript fully available?

Reviewer #1: Yes

Reviewer #2: No

4. Is the manuscript presented in an intelligible fashion and written in standard English?

Reviewer #1: Yes

Reviewer #2: No

5. Review Comments to the Author

Reviewer #1: 1. Expect the author to indicate the source of the training dataset.

2. The analysis of the images is not detailed enough.

3. The conclusion is not comprehensive and precise enough, and the third point is the future work.

Reviewer #2: 1. 4D should be given its full name at the first appearance in abstract and paper.

2. Conclusions is too long, the discussion could be addressed seperately.

3. What are the highlights in this paper？

4. The figures is not clear, the high quality should be improved.

5. The abstracts and conclusions should be improved with the highlights.

6. PLOS authors have the option to publish the peer review history of their article (what does this mean?). If published, this will include your full peer review and any attached files.

Reviewer #1: No

Reviewer #2: No

---

## [Author Response · Author response to Decision Letter 0]

9 Oct 2024

Dear Editors,

Thank you for allowing us to revise our paper, entitled “4D trajectory prediction and conflict detection in terminal areas based on an improved convolutional network” (Manuscript reference: PONE-D-24-28819). We would also like to extend our thanks to the reviewers. Their comments are very constructive and insightful for improving the quality of our paper. We have addressed all of the reviewers’ comments thoroughly. The details of our revision are provided below.

Sincerely yours.

Xin Ma, Master, Associate Professor

Linxin Zheng, Postgraduates

Xikang Lu, Postgraduates

(1)Nanjing University of Aeronautics and Astronautics, Nanjing, 211106, P. R. China;

(2)Civil Aviation Flight University of China，Guanghan, Sichuan, 618307, P. R. China.

Responses to Reviewer 1

#1. Expect the author to indicate the source of the training dataset.

Response: Thank you for your advice, this is a good question. This paper takes the ADS-B training dataset. We add the relevant description in the second section.

#2. The analysis of the images is not detailed enough.

Response: This is a very good suggestion. We supplement the substantive description of Fig. 5, Fig. 6, Fig. 7 and Fig.8 in the paper, and increase the analysis of the model and data.

#3. The conclusion is not comprehensive and precise enough, and the third point is the future work.

Response: Thank you very much for reviewers' comments. We refined the conclusion and gave a general description from the experimental data, the prediction model and the experimental results. And the third point is also changed to the description of the robustness of the article, so that the conclusion is more clear and comprehensive.

Responses to Reviewer 2

#1. 4D should be given its full name at the first appearance in abstract and paper.

Response: This is a very good suggestion. We have clearly given the full name at the first appearance in abstract and paper.

#2. Conclusions is too long, the discussion could be addressed seperately.

Response: Thank you very much for reviewers' comments. We refined the conclusion and gave a general description from the experimental data, the prediction model and the experimental results. And the third point is also changed to the description of the robustness of the article, so that the conclusion is more clear and comprehensive.

#3. What are the highlights in this paper？

Response: Thank you very much for reviewers' comments. The highlight of this paper is to create a new 4D trajectory prediction model named CNN-BiGRU, and use the trajectory distance detection function to realize the conflict assessment of the prediction results.

#4. The figures is not clear, the high quality should be improved.

Response: Thank you for your advice, this is a good question. We modify Fig.1, Fig.3 and Fig.4 respectively. After a series of fine optimization processing, the image quality has been significantly improved.

#5. The abstracts and conclusions should be improved with the highlights.

Response: Thank you very much for reviewers' comments. We refine and summarize the conclusion from three core aspects : firstly, we analyze the experimental data to ensure the solid foundation of the conclusion ; secondly, a new 4D trajectory prediction model is established to simulate the conflict detection function, and the conflict evaluation of the prediction results is realized. Finally, compared with other models, the accuracy of 4D trajectory prediction is improved.

---

## [Decision Letter · Decision Letter 1]

2 Jan 2025

4D trajectory prediction and conflict detection in terminal areas based on an improved convolutional network

PONE-D-24-28819R1

Dear Dr. linxin Zheng,

We’re pleased to inform you that your manuscript has been judged scientifically suitable for publication and will be formally accepted for publication once it meets all outstanding technical requirements.

Kind regards,

Perepi Rajarajeswari

Academic Editor

PLOS ONE

Additional Editor Comments (optional):

Reviewers' comments:

Reviewer's Responses to Questions

**Comments to the Author**

1. If the authors have adequately addressed your comments raised in a previous round of review and you feel that this manuscript is now acceptable for publication, you may indicate that here to bypass the “Comments to the Author” section, enter your conflict of interest statement in the “Confidential to Editor” section, and submit your "Accept" recommendation.

Reviewer #2: (No Response)

2. Is the manuscript technically sound, and do the data support the conclusions?

Reviewer #2: (No Response)

3. Has the statistical analysis been performed appropriately and rigorously? 

Reviewer #2: (No Response)

4. Have the authors made all data underlying the findings in their manuscript fully available?

Reviewer #2: (No Response)

5. Is the manuscript presented in an intelligible fashion and written in standard English?

Reviewer #2: (No Response)

6. Review Comments to the Author

Reviewer #2: (No Response)

7. PLOS authors have the option to publish the peer review history of their article (what does this mean?). If published, this will include your full peer review and any attached files.

Reviewer #2: No

---

## [Editor Report · Acceptance letter]

6 Jan 2025

PONE-D-24-28819R1 

PLOS ONE

Dear Dr. Zheng, 

I'm pleased to inform you that your manuscript has been deemed suitable for publication in PLOS ONE. Congratulations! Your manuscript is now being handed over to our production team.

Kind regards, 

on behalf of

Dr. Perepi Rajarajeswari 

Academic Editor

PLOS ONE